# Stress Shielding around Press-Fit Radial Head Arthroplasty: Proposal for a New Classification System Based on the Analysis of 97 Patients with a Mid-Term Follow-Up and a Review of the Literature

**DOI:** 10.3390/healthcare12030396

**Published:** 2024-02-03

**Authors:** Giuseppe Giannicola, Andrea Amura, Sebastien Prigent, Carmine Zoccali, Pasquale Sessa

**Affiliations:** 1Department of Anatomical, Histological, Forensic Medicine and Orthopedics Sciences, “Sapienza” University of Rome, Piazzale Aldo Moro 3, 00185 Rome, Italy; giuseppe.giannicola@uniroma1.it (G.G.); sebastien.prigent@uniroma1.it (S.P.); carmine.zoccali@uniroma1.it (C.Z.); 2Department of Orthopedics and Traumatology, A.O. San Camillo-Forlanini, Circonvallazione Gianicolense 87, 00100 Rome, Italy

**Keywords:** proximal radial neck resorption (PRNR), stress shielding, osteolysis, radial head arthroplasty, radial head prosthesis, RHA, implant survival, loosening, classification

## Abstract

Stress shielding (SS) around press-fit radial head arthroplasty (RHA) was recently reported as a cause of a new type of proximal radial neck resorption (PRNR). Very few studies have analyzed this phenomenon. No comprehensive classification is currently available. We thus decided to clinically and radiographically analyze 97 patients who underwent a press-fit RHA and who were followed up for a mean period of 72 months (range: 2–14 years). PRNR in the four quadrants of the radial neck was assessed. We designed a novel SS classification based on (1) the degree of resorption of the length of the radial neck and (2) the number of neck quadrants involved on the axial plane. The mean PRNR (mPRNR) was calculated as the mean resorption in the four quadrants. mPRNR was classified as mild (<3 mm), moderate (3 to 6 mm), and severe (>6 mm). Eighty-four percent of the patients presented PRNR. mPRNR was mild in 33% of the patients, moderate in 54%, and severe in 13%. In total, 6% of the patients with mild mPRNR displayed resorption in one quadrant, 18% displayed resorption in two quadrants, 4% displayed resorption in three quadrants, and 72% displayed resorption in four quadrants. All four quadrants were always involved in moderate or severe mPRNR, with no significant differences being detected between quadrants (*p* = 0.568). mPRNR has no apparent effect on the clinical results, complications, or RHA survival in the medium term. However, longer-term studies are needed to determine the effects of varying degrees of PRNR on implant failure.

## 1. Introduction

In recent decades, the widespread use of radial head arthroplasty (RHA) for the treatment of unreconstructible fractures or post-traumatic sequelae of the radial head (RH) has generated a large new cohort of patients with RH prostheses who have been followed up for a medium to long period of time [1,2,3,4]. This longer follow-up period has led to numerous RHA-related issues becoming evident: radial neck osteolysis, RHA stem subsidence and loosening, capitellar cartilage wear, fractures around the prosthesis, and PRNR around press-fit stems. PRNR, which was first described by Chanlalit et al., consists of a peculiar pattern of bone loss that is different from radiolucencies and ballooning osteolysis insofar as it starts from the periosteal portion of the radial neck and affects well-fixed stems alone [1]. The etiology of PRNR is debated in the literature; stress shielding (SS) and vascular damage of the radial head at the time of the prosthesis implant are both believed to underlie this phenomenon [1,2,3]. Giannicola et al. demonstrated that SS progresses significantly during the first 2 years, by the end of which bone resorption stabilizes definitively [2]. Very few data are currently available on SS around RHA, with no comprehensive tools existing to classify this phenomenon. Starting from these observations, we designed the present study primarily to propose a simple but comprehensive classification system of PRNR. The aim of this classification is (1) to verify whether SS severity affects either the clinical results or the risk of implant-related complications and (2) to allow the comparison of data on PRNR between future studies. The secondary aim of this study was to perform a qualitative (narrative) review of the literature on this topic.

## 2. Materials and Methods

We conducted a retrospective comparative study on 144 consecutive patients (67 males and 77 females), with a mean age of 54 years (18 to 87), who underwent RHA performed by a single elbow surgeon (G.G.) between January 2008 and December 2019.

The inclusion criteria were: (1) patients with anatomical or bipolar radial head arthroplasty consisting of conical or cylindrical short press-fit stems (Anatomical Radial Head System, Acumed, Hillsboro, OR, USA; bipolar RHS, Tornier, Montbonnot Saint Martin, France); (2) a surgical procedure performed for unreconstructible fractures or post-traumatic sequelae of the radial head (symptomatic non-union or malunion); (3) primary RHA; and (4) adult patients (>18 years). The exclusion criteria were: (1) follow-up shorter than 24 months; (2) other RHA models (i.e., monoblock prostheses, smooth stems, cemented stems, and long stems); (3) radial head fractures involving the radial neck; (4) bone tumors, congenital deformities, and bone metabolic and genetic diseases (with the exception of osteoporosis); (5) open fractures; (6) active infection; (7) revision surgery; (8) early radiological signs (within the first two years) of loosening (i.e., periprosthetic radiolucencies and ballooning osteolysis) displaying a lack of stem integration; (9) technical errors in positioning (radial overlengthening or shortening) and alignment of the implant; and (10) a lack of radiological and clinical data for the patient. 

According to the inclusion and exclusion criteria, 97 out of 144 patients were considered eligible for the present study. Intraoperative diagnoses were terrible triad in 29 cases, unreconstructible radial head fractures in 21 cases, fracture–dislocation of the proximal ulna and radius in 19 cases, unreconstructible radial head fractures associated with elbow dislocation in 6 cases, non-union in 4 cases, chronic complex persistent elbow instability in 10 cases, and post-traumatic stiffness in 8 cases.

Informed consent was provided by each participant, in accordance with the Declaration of Helsinki. Ethics committee approval was obtained from Sapienza University of Rome (prot.0564/2022, rif.6773).

In view of the self-limiting nature of PRNR, the X-rays performed in the immediate post-operative period (within 2 weeks of surgery) and after a minimum of 24 months were considered for the purposes of this study [2]. The radiological views used to evaluate the four quadrants of the radial neck in all of the patients were, as described by Chanlalit et al. [1] and Giannicola et al. [5]: standard anteroposterior (AP) view (elbow in full extension with the forearm supinated) and standard lateral (L) view (elbow flexed at 90° with the forearm supinated). Radiographs in neutral or pronated rotation in both the AP and L views were also obtained in patients with rotatory stiffness; in patients with elbow stiffness affecting extension, the AP view was performed with the beam perpendicular to the forearm. Radiographs were then digitized and analyzed by means of dedicated software that normalized the proportions (OsiriX^®^ version 10.0, 64-bit; Pixmeo, Geneva, Switzerland). A high-resolution monitor was used. Two authors performed each of the measurements in the study separately. The mean value between the two measurements was used for the statistical analysis. A third measurement was performed jointly by both operators if there were >1 mm differences between their measurements. 

The radial neck was divided into four quadrants according to the standard anatomical position of the human body: the anterior quadrant (AQ), the posterior quadrant (PQ), the medial quadrant (MQ), and the lateral quadrant (LQ) in order to calculate the degree of PRNR. The LQ and MQ were identified in the AP view, while the L view was used to identify the AQ and PQ (Figure 1). The post-operative radiographs were used to evaluate the initial distance between the collar of the prosthetic stem and the cortex of each quadrant. At the last follow-up, PRNR was measured in each quadrant by quantifying the distance between the collar of the prosthetic stem and the endosteal portion of the cortex in each quadrant, since this was the portion that was affected least by PRNR. Subsequently, the mean PRNR (mPRNR) for each patient was calculated by evaluating the arithmetic mean value of the four quadrants. 

At the last follow-up, the mPRNR was classified into 3 different grades: mild (<3 mm resorption), moderate (3–6 mm resorption), and severe (>6 mm resorption) (Figure 2). The 3 mm cut-off was chosen on the basis of the mean total length of the radial neck (normally from 8 to 9 mm); in particular, we considered the anatomical distance between the lower margin of the radial head articular surface on its ulnar side, which corresponds to the lesser sigmoid notch, and the proximal margin of the bicipital tuberosity of the radius [6,7]. The number of neck quadrants involved for each mPRNR grade was recorded.

For the clinical evaluation, the Mayo Elbow Performance Score (MEPS), the patient—American Shoulder and Hand Score (p–ASES–e), and the Disability of the Arm, Shoulder and Hand score (q–DASH) were calculated at the last follow-up [8,9,10]. The range of motion (ROM) in extension–flexion (E–F) and in pronation–supination (P–S) was also recorded. The radiographic signs of RHA-related complications such as late radiolucencies and periprosthetic osteolysis, heterotopic ossifications (HOs), aseptic loosening, and mechanical failure (i.e., disassembly and fractures of the implant) were investigated. Articles on PRNR in the literature were reviewed.

### Statistical Analysis

The reliability of the measurement method was assessed in a preliminary study conducted on 25 radiographs. Two independent observers performed all of the assessments. Measurements were repeated twice (time 1 and time 2) at an interval of three weeks; the order of the radiographs was randomly changed at time 2 to generate a new sequence. The intraclass correlation coefficient (ICC) was used to evaluate the interobserver and intraobserver reliability. The ICC analysis yielded a high coefficient for both the interobserver and intraobserver reliability (k > 0.80 in each type of measurement performed). The mean and standard deviations were calculated for each numerical variable range. The ANOVA test for repeated measures with Tukey’s post hoc analysis was used to assess any differences within quadrants and between quadrants in each patient at each follow-up. The one-way ANOVA test with Tukey’s post hoc analysis was used to compare the clinical data between subgroups stratified according to PRNR severity. The Spearman correlation (rho) was used to assess any correlations between the mPRNR and the clinical variables examined (MEPS, DASH, p-ASES, and ROM). The significance level was set at *p* = 0.05. SPSS (ver.21, IBM, Chicago, IL, USA) was used for the statistical analysis.

## 3. Results

The mean follow-up of the 97 patients was 72 months (range: 24 to 166). In total, 18 of the 97 patients were followed up for >9 years, while 51 were followed up for between 4 and 8 years. At the final follow-up, 82 of the 97 patients (84%) displayed radiographic signs of PRNR in at least one of the four quadrants; the mean PRNR was 3.7 mm (range: 0–10.9) in the anterior quadrant, 3.2 mm (range: 0–9.2) in the posterior quadrant, 3.5 mm (range: 0–10.7) in the lateral quadrant, and 2.8 mm (range: 0–8.4) in the medial quadrant. No significant differences in PRNR were observed between quadrants at the last follow-up (*p* = 0.568).

PRNR was mild in 27 patients (33%), who had an mPRNR of 1.9 mm (range: 0–2.9); PRNR was moderate in 44 patients (54%), who had an mPRNR of 4.5 mm (range: 3.1–5.9); lastly, PRNR was severe in 11 patients (13%), who had an mPRNR of 7 mm (range: 6–9.4). The typical progression of PRNR over time for an anatomical RHA is shown in Figure 3, while that for a bipolar implant is shown in Figure 4. Table 1 shows the clinical result scores in the subgroups stratified according to PRNR severity. No significant differences were observed in either the clinical characteristics or the clinical scores between the three subgroups. Indeed, no significant correlations (*p* = 1) (Table 2) were detected between the degree of PRNR and any of the clinical parameters in each subgroup.

When the number of radial neck quadrants involved in PRNR was considered, 6% of the 27 patients with mild PRNR displayed resorption in one quadrant, 18% displayed resorption in two quadrants, 4% displayed resorption in three quadrants, and 72% displayed resorption in four quadrants. All four quadrants were instead always involved in all of the patients with moderate or severe mPRNR. A significant difference in the number of quadrants involved was detected between the mild PRNR group and the moderate/severe PRNR groups (*p* = 0.044).

At the final follow-up, 23 patients displayed HOs around the radial neck. According to Hasting and Graham’s classification, we observed type I, type IIB, type IIC, and type IIIB HOs in 14, 4, 3, and 2 patients, respectively. The prevalence of HOs in the group of patients affected by PRNR was 14% (12 cases), while the same prevalence was 73% (11 cases) in the group without PRNR. In the latter group, HOs were classified as type I, type IIB, type IIC, and type IIIB in five, three, two, and one case(s), respectively. In other words, in the group of patients without PRNR, 6 out of 11 (54%) displayed symptomatic HOs while in the group of patients with PRNR, only 3 out of 12 (25%) displayed symptomatic HOs. Among the 12 patients who displayed PRNR and HOs, the mPRNR was mild in 7 cases and moderate in 5. 

No implant-related complications were detected at the last radiographic evaluation. No patients underwent reintervention during the follow-up owing to stem failure-related complications. Three patients suffered a disassembly of a bipolar prosthesis 4, 5, and 10 years after the index procedure following a new trauma. In the first patient, the radial head component was replaced and the LCL was reconstructed. In the second and third patients, the disassembly was associated with capitellar fracture; these two cases both underwent osteosynthesis with two headless screws and replacement of the radial head component. The mPRNR was moderate in one of these three patients, severe in another, and moderate in the last. The stem appeared to be well integrated in all three cases, with no need for revision (Figure 5).

The results of the review of the literature are shown in Table 3.

## 4. Discussion

The aim of this study was to propose a new quantitative and simple classification of PRNR so as to allow researchers to compare the results from different studies on SS around RHA. We clinically and radiographically analyzed a large series of patients who underwent a press-fit RHA. The novel classification of SS is based on the degree of resorption of the length of the radial neck and the number of neck quadrants involved in the axial plane. The mPRNR was classified as mild (<3 mm), moderate (3 to 6 mm), and severe (>6 mm); approximately 84% of the patients presented PRNR in at least one of the four quadrants. The mPRNR was mild, moderate, and severe in 33%, 54%, and 13% of the patients, respectively. In mild mPRNR, one or more quadrants were involved, whereas in moderate and severe mPRNR, all four quadrants were always affected. PRNR had no apparent effect on the clinical results or RHA survival in the medium term. However, further long-term follow-up studies are warranted to clarify the reasons underlying the failure of stems in patients affected by different grades of bone resorption. 

PRNR is a well-known, though as of yet not fully understood phenomenon of bone resorption around well-fixed radial head press-fit stems. There has recently been growing interest in the literature about this topic, mainly owing to the availability of larger cohorts of patients with RHA with mid- to long-term follow-up. Concerns have arisen regarding the possible link between PRNR and reduced implant survival or higher complication rates. In their study on 26 patients who received one of five different RH arthroplasty designs, Chanlalit et al. [1] reported that PRNR occurred in 65% of their patients at 33 months of follow-up, with no progression in the phenomenon being observed beyond the 26th month after the surgical procedure. Levy et al. [3] reported that PRNR occurred in 67% of the patients in their series, who received an anatomical press-fit RHA and were followed up for 30 months. Giannicola et al. [2] reported that PRNR occurred in 84% of their patients, who received bipolar and anatomical press-fit stems and had a mean follow-up of 6 years, with no further progression of this phenomenon being observed beyond the 24th month after the surgical procedure. Lastly, Lee et al. [4] reported a 40% prevalence of PRNR in their series of 35 patients, who received either anatomical press-fit RHA or bipolar cemented RHA and had a mean follow-up of 45 months. PRNR, hence, emerges as a common radiological finding in the first two years after an RH replacement, with a self-limiting nature but uncertain clinical behavior in the very long term [2].

Currently, the most credited etiopathogenesis for PRNR is SS, as suggested by Chanlalit et al. [1]. In the study by Lee et al. [4], the authors reported a significantly higher prevalence of PRNR in anatomical press-fit stems than in bipolar cemented stems (63% vs. 13%). The finding of a reduced incidence of SS in cemented arthroplasty is consistent with previous reports in the literature on cemented RHA [11] and hip arthroplasty [12,13,14,15], thus potentially validating the mechanical origin of PRNR. On the other hand, some observations raise doubts about the possibility that other factors may contribute to the origin of PRNR. When the relationship between the design of the prosthetic stems and PRNR was assessed in studies in the literature, no differences were detected between the different press-fit stem shapes (conical vs. cylindrical), prosthetic designs (bipolar, anatomical, and mono-block), or stem fixation techniques (press-fit and cemented). In the study by Chanlalit et al. [1], in which five different types of RHA were considered, no differences emerged in the incidence of PRNR between implants with different characteristics. However, the small sample size might have prevented the authors from finding significant correlations. In the study by Giannicola et al. [5], the authors analyzed two radial head implants to assess whether the shape and the size of the stems affected PRNR. The degree of resorption in the four quadrants of the radial neck in a bipolar implant with a conical stem and an anatomical implant with a cylindrical stem were analyzed. Furthermore, the ratios between the medullary canal and prosthetic stem dimensions at three different levels of the stems were calculated to analyze the correlations between the degree and the type of canal filling of the stems and PRNR. They observed that the shape and size of the stems were not correlated with the degree of PRNR despite a significant difference in the canal filling between the two implants, thereby casting doubts on a possible biomechanical explanation for this phenomenon. The authors suggested that damage to vascularization in the radial neck during surgery may represent one of the causes contributing to PRNR. This hypothesis may be supported by the fact that we observed that PRNR was often absent in patients with heterotopic ossifications around the radial neck. In particular, we observed HOs in 73% of patients without PRNR, while the same condition was measured at 14% in patients with PRNR. As is well known, heterotopic bone is a richly vascularized tissue and this might prevent bone resorption; the onset of PRNR was never detected after HO maturation. If the initial injury pattern is considered, no correlations were found with the onset of PRNR [1,2,3] in any of the studies with the exception of the study of Lee et al. [4], who reported a significant correlation between the PRNR rate and bilateral ligamentous injury of the elbow. The small number of patients considered represents the main limitation of their study. The fact that no explanation for this finding was provided by the authors further suggests that it might have been a chance occurrence. 

In addition to providing data on the mean PRNR observed in their series [1,2,3,4] and the related clinical results, two of the authors of the aforementioned studies also proposed a scheme for PRNR severity stratification. Chanlalit et al. [1] designed a classification that is based on qualitative criteria: stages are divided into cortical bone thinning (stage I), a partially exposed stem (stage IIa), a circumferentially exposed stem (stage IIb), and mechanical or impending failure (stage III). This classification does not include any quantitative parameters to stratify PRNR, such as the extension of bone resorption in terms of the radial neck length. Moreover, the authors used a 3D reconstruction designed by dedicated software to correctly assess the circumferential involvement of the radial neck, which might not always be feasible in clinical practice. A similar qualitative classification system was proposed more recently by Lee et al. [4]. PRNR was graded into three types according to the lateral radiographic images: bone resorption just beneath the prosthetic collar (Type 1); bone resorption that progressed to the midportion of the prosthetic stem without interfering with stability (Type 2); and more distal bone resorption which predicted instability of the prosthesis (Type 3). Lee et al. did not find any type 3 PRNR in their series. This classification, which is based exclusively on the lateral radiographs, might not be able to correctly evaluate PRNR in the lateral and medial quadrants; the qualitative nature of this classification may also represent a limitation insofar as it might not be reproducible enough to allow an objective comparison between studies. 

Given these premises, the aim of the present study was to propose a simple quantitative PRNR classification to fill this gap in the current literature. The present study was based on 97 patients who underwent RHA with short press-fit stems and had a mean follow-up of 6 years; approximately 82 of these 97 patients (84%) displayed some degree of PRNR. The patients with PRNR were then stratified into three subgroups (mild, moderate, and severe) according to the degree of PRNR. We considered an mPRNR <3 mm as mild, an mPRNR between 3 and 6 mm as moderate, and an mPRNR > 6 mm as severe. The 3 mm cut-off was chosen on the basis of the anatomical distance (normally ranging from 8 to 9 mm [7,8]) between the lower margin of the radial head articular surface on its ulnar side, which corresponds to the lesser sigmoid notch, and the proximal margin of the bicipital tuberosity (i.e., the length of the radial neck). According to the data in the present study, 54% of the sample presented a moderate mPRNR value (3–6 mm) and 33% presented a mild mPRNR value. A severe mPRNR was observed in only 13% of the sample, which indicates that more severe forms of PRNR are relatively rare. When the number of radial neck quadrants involved in PRNR was considered, 6% of the patients with mild mPRNR displayed resorption in one quadrant, 18% displayed resorption in two quadrants, 4% displayed resorption in three quadrants, and 72% displayed resorption in four quadrants. All four quadrants were instead always involved in all of the patients with moderate or severe mPRNR, which highlights the circumferential nature of PRNR in all but the lower grades. 

The correlation between PRNR severity and clinical scores or complication rates has been addressed in a few previous studies [1,3,4], and none of those studies included a medium to long follow-up period or were based on a large cohort of patients. In this study, no significant correlation between the severity of PRNR and the clinical results was found in the medium term, nor were higher complication rates observed in patients with more severe signs of PRNR. These findings appear to confirm the fact that the severity of PRNR has no impact in the clinical setting, as is also revealed by the excellent clinical scores reported even in more severe cases of PRNR. This finding is in keeping with what is currently known about the more widely studied phenomenon of SS in total hip replacement, for which longer follow-up data are available for different implant designs; there is no evidence that clinical outcomes are related to bone remodeling regardless of the stem design [16,17]. The complication rate in the present study is in line with other reports in the literature and is not related to the stem design. 

According to what we currently know, we may hypothesize that when the length of the prosthetic stem within the bone decreases as a result of PRNR, changes in micromotion exert higher stress on the stem tip, which may modify the implant–bone interface and increase the risk of aseptic loosening. In their cadaveric model using a 25 mm fixed-length RH prosthesis, Shukla et al. [18] measured stem tip micromotion after progressively resecting larger amounts of the radial neck. They introduced the concept of the cantilever quotient (CQ), which is the ratio between the resected amount of bone and the overall length of the prosthesis, which includes the radial head, neck, and stem. Implants with a CQ greater than 0.35 are believed to be at risk of loosening owing to the increased stem micromotions and the lack of stem integration. In particular, Shukla et al. demonstrated that when the length of the prosthetic stem within the bone was reduced and the amount of bone resection was increased, which further exposed the prosthetic stem above the radial neck cut, the micromotions dramatically increased at the stem tip, thereby resulting in a concrete risk of stem loosening. In their retrospective study on 65 patients with 4 different types of RH prostheses. Laumoniere et al. [19] reported that in the presence of a favorable micromotion environment, the incidence of aseptic loosening was lower when longer stems (30 mm) were used. Lastly, in a biomechanical study by Moon et al. [20], two groups of eight human cadaveric radii were implanted with custom titanium sand-blasted cylindrical press-fit RHA with an 8 or 9 mm diameter at two different resection levels of the radial head (10 mm and 21 mm from the radial head articular rim in the first and second group, respectively). Afterward, 5 mm at a time were added to the initial stem length of 10 mm until the maximum stem length of 30 mm was attained. Each stem length increase was followed by a micromotion test using a dedicated device. The authors demonstrated that when longer stems (>25 mm) were used, micromotion was significantly reduced in the group with a 10 mm resection level; in contrast, no significant differences were observed in micromotion between the 5 different stem lengths in the group with a 21 mm resection level. Moon et al. postulated that this was due to the radius isthmus being located at a mean distance of 13 mm from the radial head–neck junction; in large resections, the prosthetic stems might fit into this restricted portion of the radial canal more tightly, thereby providing a stiffer press-fit with reduced micromotion. 

These three studies thus highlight the close correlation between the length of the stems, the length and shape of the radial neck, and the stability and integration of the implants. While these studies focused on the effect of micromotion on the initial stage of the bone integration of press-fit stems, the “reverse phenomenon”, i.e., the effect of neck narrowing and shortening (caused by PRNR) on well-integrated stems, has yet to be thoroughly investigated. Our findings appear to indicate that PRNR neither affects stem stability nor reduces the mechanical resistance of the radial neck to new traumatic events, at least in the medium term. This is demonstrated by the fact that the stems in all three patients who underwent reinterventions following new traumas were stable intraoperatively. However, these few cases are not enough to exclude PRNR as a risk factor for periprosthetic fractures or aseptic loosening.

Although PRNR has been described as an asymptomatic phenomenon that does not affect stem survival in the medium term, very long-term follow-up studies may be needed to witness the failure of press-fit stems in a significant number of patients and, consequently, to analyze and understand the reasons underlying any such failures. For these reasons, we expect the classification we propose in this study to assume a prognostic value when larger numbers of mechanical failures in patients with more severe PRNR emerge in the longer term. In addition, the fact that we only considered two types of press-fit stem designs in this study prevented us from extending the results on the correlation between PRNR severity and clinical results or complication rates to other stem designs or stem fixation methods (cemented implants). The use of this classification for different RHA designs (i.e., cemented implants, loose-fit implants, or different stem designs) will enhance its prognostic value by potentially revealing any failure patterns that might be associated with the various prosthetic implant designs available. Moreover, the present classification may also prove useful in the design of new radial head prosthetic implants aimed at reducing the onset and severity of PRNR due to SS, should the biomechanical nature of PRNR be confirmed. Moreover, patients with more severe forms of PRNR may benefit from closer follow-up visits in order to promptly identify potential prosthetic failure and receive the appropriate treatment. Lastly, this classification may serve as a basis for a therapeutic algorithm for RHA revision surgeries thanks to the light it sheds on different grades of radial neck bone loss. 

## 5. Conclusions

This study proposes a new quantitative classification of PRNR after RHA that is meant to offer the possibility to objectively classify this phenomenon and to readily exchange data on treated patients. Although PRNR is currently considered a self-limiting asymptomatic radiological finding, very long-term follow-ups may challenge the claim that PRNR has no detrimental effect on prosthetic survival by revealing a different behavioral pattern in more severe cases. For these reasons, a simple but complete tool is needed. 

## Figures and Tables

**Figure 1 healthcare-12-00396-f001:**
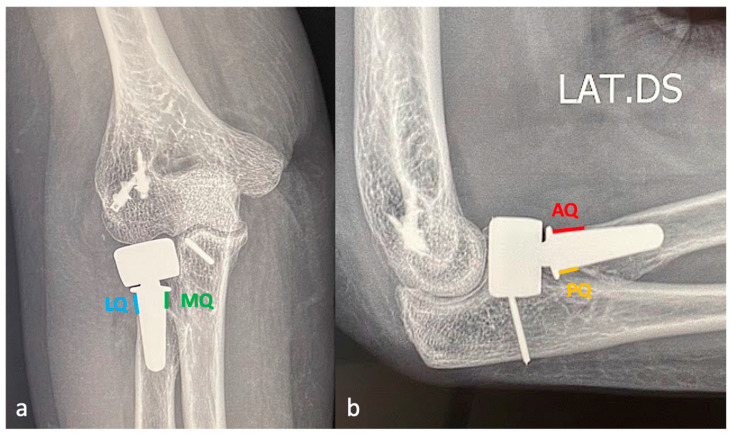
(**a**) The anterior–posterior view with the forearm supinated and the elbow in full extension allows the lateral (LQ) and medial quadrants (MQ) of the radial neck to be identified; (**b**) the lateral view with supinated forearm and the elbow flexed at 90° allows the anterior (AQ) and posterior quadrants (PQ) to be identified.

**Figure 2 healthcare-12-00396-f002:**
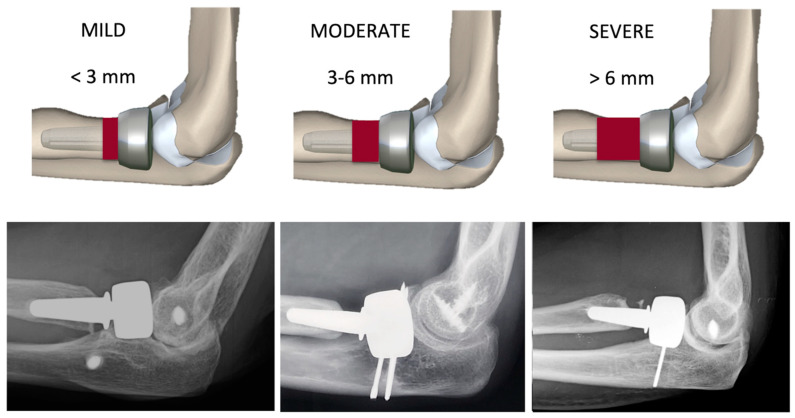
Classification of proximal radial neck resorption (PRNR). The degree of PRNR was classified into three grades: mild (PRNR < 3 mm), moderate (PRNR 3–6 mm), and severe (PRNR > 6 mm).

**Figure 3 healthcare-12-00396-f003:**
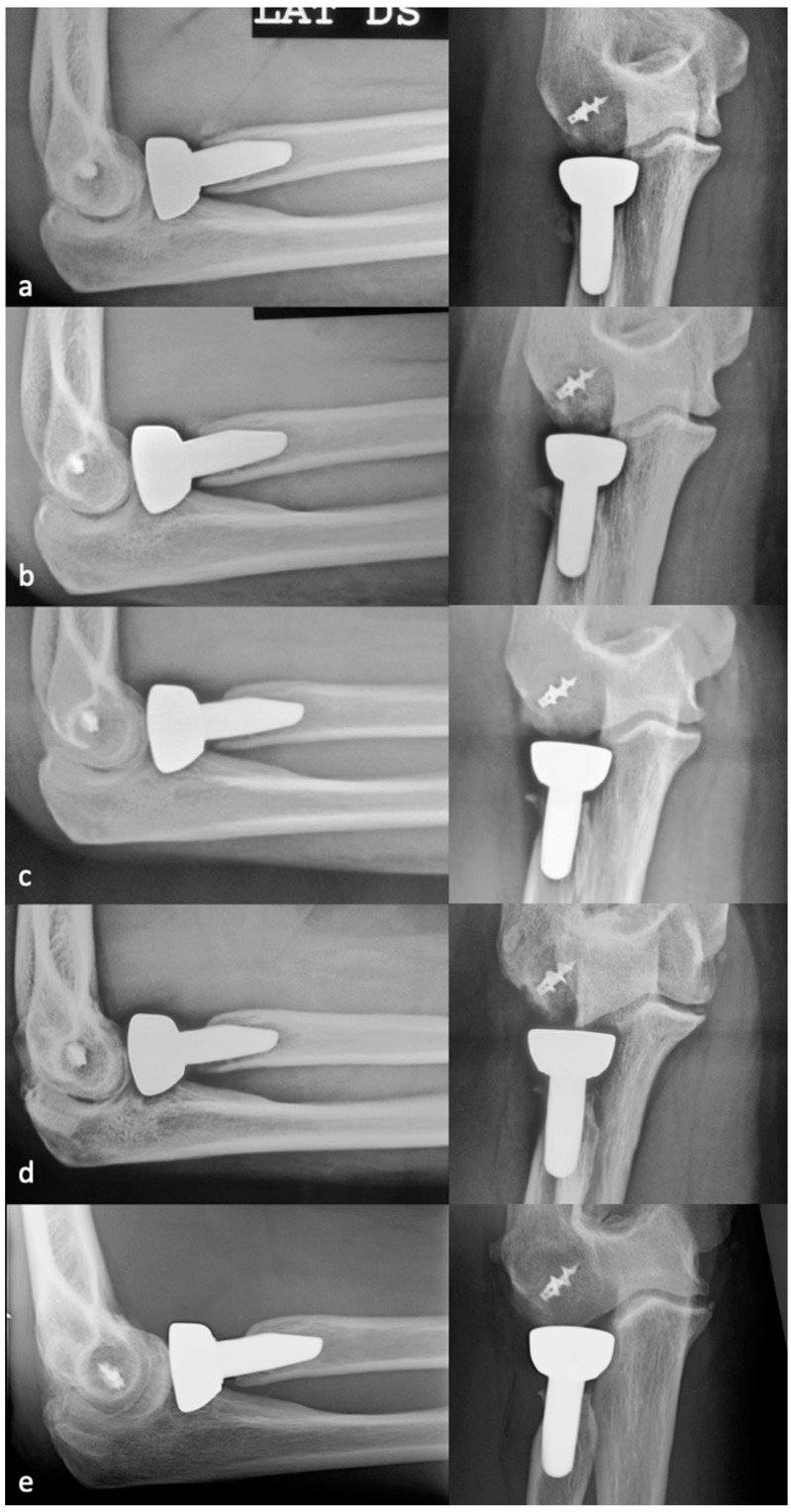
Progression of PRNR over time in an anatomical press-fit RHA. X-rays performed postoperatively at 2 weeks (**a**), 6 months (**b**), 1 year (**c**), 2 years (**d**), and 7 years (**e**). PRNR progressed until the end of the second year, while no further progression was observed beyond 2 years.

**Figure 4 healthcare-12-00396-f004:**
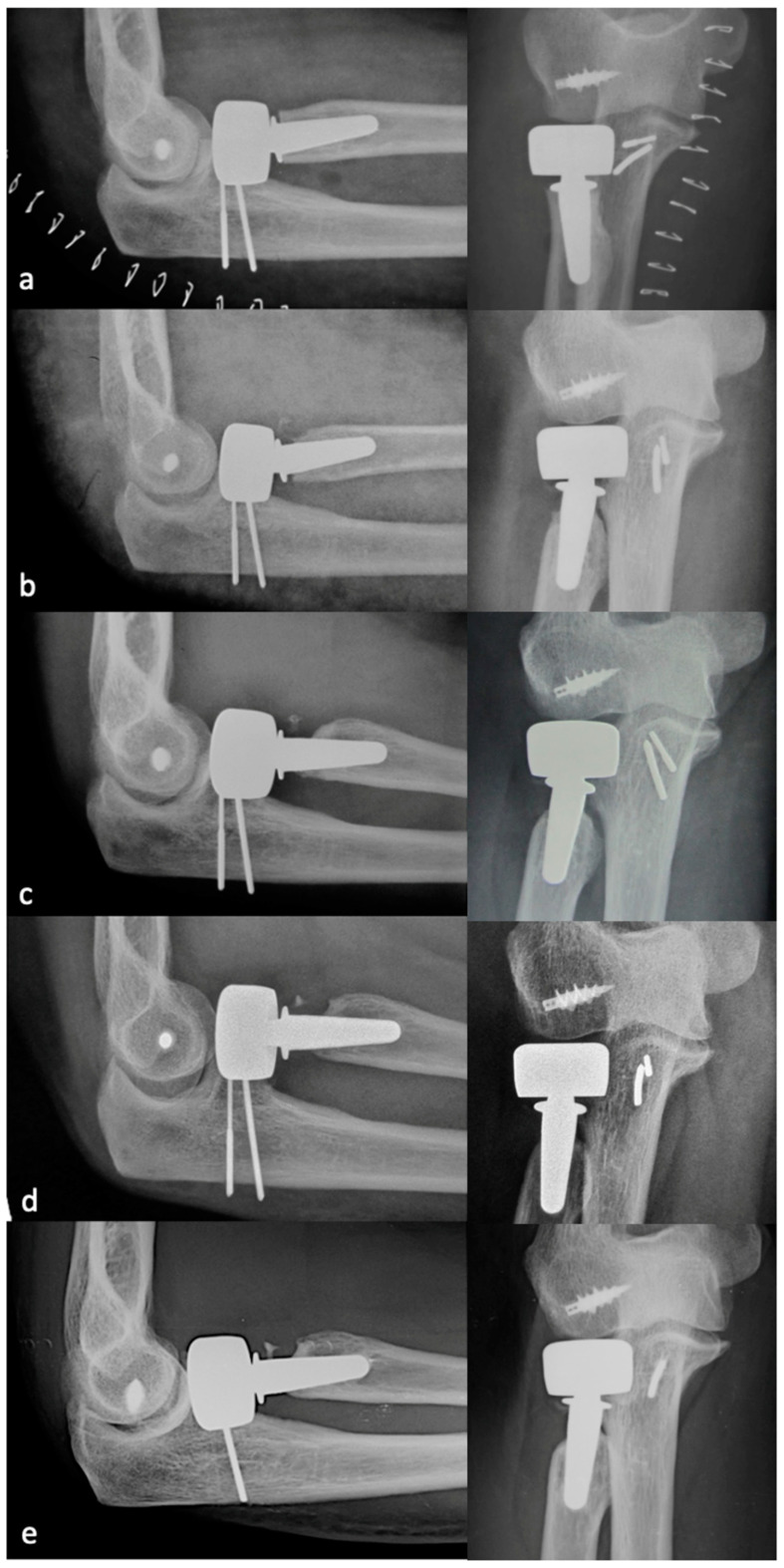
Progression of PRNR over time in a bipolar press-fit RHA. X-rays performed postoperatively at 2 weeks (**a**), 6 months (**b**), 1 year (**c**), 2 years (**d**), and 9 years (**e**). PRNR progressed until the end of the second year, while no further progression was observed beyond 2 years.

**Figure 5 healthcare-12-00396-f005:**
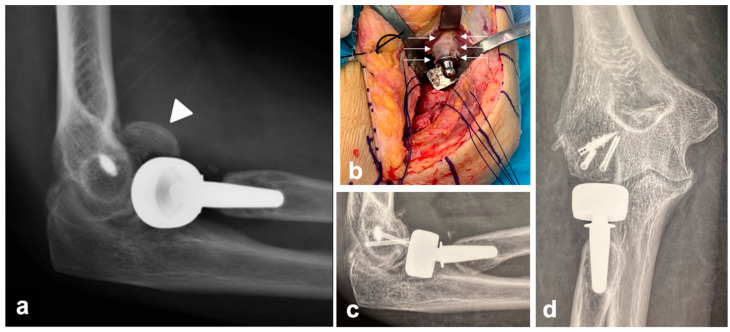
Clinical case of a patient who experienced disassembly of a bipolar prosthesis 10 years after the index procedure following a new trauma. The disassembly was associated with a capitellum fracture (white arrowhead) (**a**); the mPRNR in this patient was moderate. The stem appeared to be well integrated, with no need for stem revision (**b**). The intraoperative appearance of PRNR is also shown in (**b**) (white arrows). Osteosynthesis with two headless screws and replacement of the radial head component were performed (**c**,**d**).

**Table 1 healthcare-12-00396-t001:** Clinical data, functional scores, and ROM in the subgroups stratified according to PRNR severity.

	Mild (n = 27)	Moderate (n = 44)	Severe (n = 11)	*p*=
Sex M F	9 18	16 28	5 6	
Age	54.17 (13.8)	53.4 (14.8)	51.3 (11.8)	0.883
Follow-up	52.27 (41.1)	64.6 (34.2)	48 (44.8)	0.390
MEPS	96.33 (7.4)	91.5 (18)	98.7 (2.3)	0.267
Q-DASH	4.9 (14.1)	9.9 (18.4)	0.29 (0.8)	0.240
P-ASES	51.94 (5.6)	51.5 (4.3)	50.5 (1.4)	0.734
SUP (°)	81.33 (15.4)	80.8 (11.1)	70 (29.7)	0.159
PRON (°)	81.33 (16.9)	80.9 (11.5)	76.8 (31.2)	0.778
EXT (°)	18.33 (16.6)	12.6 (11.6)	16.25 (14.8)	0.618
FLEX (°)	137.3 (11.9)	135.6 (10.3)	133.1 (9.6)	0.184

Values reported refer to mean and standard deviation (SD); PRNR: Proximal Radial Neck Resorption; MEPS: Mayo Elbow Performance Score; Q-DASH: quick Disability Arm and Shoulder; P-ASES: American Shoulder and Elbow Surgery score; sup: supination; pron: pronation; ext: extension; flex: flexion.

**Table 2 healthcare-12-00396-t002:** Correlation between mPRNR and clinical variables in the subgroups stratified according to PRNR severity.

	Mild (n = 27)	Moderate (n = 44)	Severe (n = 11)
Age	*p* = 0.854 Rho = 0.037	*p* = 0.668 Rho = −0.074	*p* = 0.233 Rho = 0.476
Follow-up	*p* = 0.608 Rho = 0.103	*p* = 0.323 Rho = 0.165	*p* = 0.629 Rho = −0.204
MEPS	*p* = 0.210 Rho = 0.249	*p* = 0.522 Rho = 0.107	*p* = 0.547 Rho = 0.252
Q-DASH	*p* = 0.490 Rho = −0.139	*p* = 0.655 Rho = 0.075	*p* = 0.310 Rho = 0.412
P-ASES	*p* = 0.66 Rho = −0.089	*p* = 0.283 Rho = 0.179	*p* = 0.846 Rho = −0.082
SUP (°)	*p* = 0.264 Rho = 0.223	*p* = 0.927 Rho = −0.015	*p* = 0.550 Rho = −0.250
PRON (°)	*p* = 0.429 Rho = 0.159	*p* = 0.128 Rho = 0.251	*p* = 0.949 Rho = −0.027
EXT (°)	*p* = 0.084 Rho = −0.339	*p* = 0.510 Rho = −0.110	*p* = 0.520 Rho = −0.268
FLEX (°)	*p* = 0.238 Rho = 0.235	*p* = 0.899 Rho = −0.021	*p* = 0.401 Rho = 0.346

mPRNR: mean proximal radial neck resorption; MEPS: Mayo Elbow Performance Score; Q-DASH: quick Disability Arm and Shoulder; P-ASES: American Shoulder and Elbow Surgery score; sup: supination; pron: pronation; ext: extension; flex: flexion.

**Table 3 healthcare-12-00396-t003:** Literature review.

Author and Year (No of Implant Types)	No. of Patients	Patients with PRNR	Mean Follow-Up	Clinical Results	Implant-Related Reintervention	Complications
Chanlalit et al., 2012 [1] (5 RHA types)	26	17 (63%)	33 months	NA	NA	NA
Levy et al., 2016 [3] (1 RHA type)	15	6 (40%)	30 months	M-ASES 70; MEPS 85; VAS 2; SANE 75	0	6 loosening
Giannicola et al., 2021 [5] (2 RHA types)	52	50 (96%)	33 months	NA	NA	NA
Lee et al., 2023 [4] (2 RHA types)	35	14 (40%)	34 months	MEPS 77.8; q-DASH 23.8, VAS 0.8	NA	NA
Giannicola et al., 2023 [2] (2 RHA types)	97	82 (84%)	72 months	MEPS 94.3, p-ASES 93.6, DASH 6.7	3 disassembly due to new trauma	6 symptomatic stiffness, 9 ulnar neuropathy, and 31 asymptomatic PL instability

NA: not assessed; VAS: visual analogical scale; MEPS: Mayo Elbow Performance Score; Q-DASH: quick Disability Arm and Shoulder; p-ASES: patient—American Shoulder and Elbow Surgery score; SANE: single assessment numeric evaluation.

## Data Availability

Data are contained within the article.

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
