# Peer review of "Stress Shielding around Press-Fit Radial Head Arthroplasty: Proposal for a New Classification System Based on the Analysis of 97 Patients with a Mid-Term Follow-Up and a Review of the Literature"

_healthcare, 2024, doi:10.3390/healthcare12030396_

Round 1

Reviewer 1 Report

Comments and Suggestions for Authors

The paper titled "Stress-Shielding Around Press-Fit Radial Head Arthroplasty: Proposal For A New Classification System Based On The Analysis Of 97 Patients With A Mid-Term Follow-Up And A Review Of The Literature" by Giuseppe Giannicola et al. provides a comprehensive analysis of stress-shielding (SS) around press-fit radial head arthroplasty (RHA). The authors aim to address the lack of a comprehensive classification system for this phenomenon.

Despite minor issues, the manuscript is well-written and scientifically sound. The proposed SS classification system is a valuable contribution to the field. I recommend minor revisions to address the noted spelling and typographical errors for enhanced clarity. Additionally, the authors may consider providing further discussion on the potential clinical implications of their findings and the limitations of the study.

1. In materials & methods you state "We conducted a retrospective comparative study on a prospective database contain-57 ing 144 consecutive patients (67 males and 77 females), with a mean age of 54 years (18 to 58 87), who underwent RHA performed by a single elbow surgeon (GG) between January 59 2008 and December 2019. " - Please erase 'on a prospective database' since it is a retrospective study. 

2. Please write 1-2 phrases in the discussion or conclusion what your findings bring as consequences for the clinical practice now. 

Thank you very much for this interesting work! All the best to you and your further research

Author Response

1.In materials & methods you state "We conducted a retrospective comparative study on a prospective database contain-57 ing 144 consecutive patients (67 males and 77 females), with a mean age of 54 years (18 to 58 87), who underwent RHA performed by a single elbow surgeon (GG) between January 59 2008 and December 2019. " - Please erase 'on a prospective database' since it is a retrospective study. 

R: We thank the reviewer for the valuable comments. We have modified the sentence as required and changed it to: "We conducted a retrospective comparative study on 144 consecutive patients (67 males and 77 females), with a mean age of 54 years (18 to 87), who underwent RHA performed by a single elbow surgeon (GG) between January 2008 and December 2019." (lines 57-59)

2. Please write 1-2 phrases in the discussion or conclusion what your findings bring as consequences for the clinical practice now. 

R: According to the reviewer’s suggestion, we have added the following sentence in the discussion section: "Besides, patients with more severe forms of PRNR may benefit from closer follow-up visits in order to promptly identify potential prosthetic failure and receive appropriate treatment." (lines 429-431).

Reviewer 2 Report

Comments and Suggestions for Authors

Dear Authors,

A really novel topic is highlighted in this study. Some really important conclusions have been demonstrated. However, there are some points regarding methods and results needing further clarification. 

Line 62: Giannicola et al. (2021) has proved that proximal radial neck resorption (PRNR) is independent of stem's shape and size. Please explain the rationale of using as inclusion criterion only conical or cylindrical short press-fit stems.

Lines 64-65: Please clarify the meaning of "post-traumatic sequelae of the radial head"? Is there any possibility axis malrotation or early post-traumatic arthritis affect in an undefined fashion the stability and biomechanics of radio-capitulum joint?

Line 65: Please provide possible causes and conditions leading to primary radial head arthroplasty. In which way did these conditions influence joint biomechanics and possibly stress shielding?

Line 70: Please define in time manner the meaning of early. Please be also exact about the types of loosening.

Lines 71-72: Please define these technical errors.

Line 74: Please describe the surgical management of these 29 cases of elbow terrible triad. How were the soft tissue injuries (tendons, ligaments etc.) assessed? In case of different management , did PRNR show any difference in severity or extension? Please mention these in the manuscript to prove the lack of heterogeneity in your sample or, otherwise, form subgroups excluding statistical errors.

Lines 76-77: Please mention the surgical management of the elbow dislocation apart of radial head reconstruction. As previously mentioned, the target is to prove that your sample has low or no heterogeneity.

Lines 77-78: Please mention causes and degrees of elbow instability. How was this managed in each case? In which way did affect this the elbow biomechanics?

Line 78: Please mention causes and degrees of stiffness. How was this managed in each case? Which was the early postoperative rehabilitation protocol and which was the variance of range of motion? Could a limited range of motion affect bone density of radial head? 

Line 83-84: As you mentioned the least time frame for radiological assessment was 2 years. During this period, were no x-rays taken to rule out any other causes than stress shielding? Please clarify.

Line 89: What did you mean rotatory stiffness? Is there a special subgroup of previous mentioned 8 cases of post-traumatic stiffness? Please clarify.

Line 96: Please mention exactly what is the meaning of pronounced differences. Moreover, was this range predefined or was assessed during the measurements?

Lines 114-118: The rationale of cut-off points for proposed classification was fully and adequately presented. This is mandatory for any newly introduced classification.

Lines 130-131: Please mention in details the flowchart of this network review. Was it systematic review? If yes present it following PRISMA guidelines. In any other case you have to conduct a systematic review where pool data will be formed (if heterogeneity is low) and quantitive synthesis will be performed.

Line 148: Regarding your results, you have to proof that your sample has no or minor heterogeneity and different causes (injuries or conditions) leading to radial head arthroplasty do not influence in different manner neck resorption. Quite recently, Giannicola et al (2023) has published the same results. Apart from the newly introduced classification, what is novel in this manuscript?

Line 165: Please clarify what is the meaning of brackets in age, follow-up, MEPS, Q-DASH, P-ASES, SUP, PRON,EXT, FLEX.

Line 212: Discussion section will be reassess after revisions and modification made in the previous sections according to above comments.

Author Response

Line 62: Giannicola et al. (2021) has proved that proximal radial neck resorption (PRNR) is independent of stem's shape and size. Please explain the rationale of using as inclusion criterion only conical or cylindrical short press-fit stems.

R: We included only conical and cylindrical press-fit stems as these were the implants that we have been using during the past 15 years, and these represent, as well, the two most studied implants in literature. Since loose-fit implants have been proven not to cause PRNR, they were not included in this study. For similar reasons, we also excluded cemented implants, as these have shown to cause a lesser degree of PRNR (Lee et al. reference [4], lines 271-273). On the bases of this evidence, we deemed it more correct to focus our research on press-fit stems.

Lines 64-65: Please clarify the meaning of "post-traumatic sequelae of the radial head"? Is there any possibility axis malrotation or early post-traumatic arthritis affect in an undefined fashion the stability and biomechanics of radio-capitulum joint?

R: We thank the reviewer for the observation. We have added in the manuscript that post-traumatic sequelae are symptomatic malunions and non-unions (pain and stiffness) requiring a radial head arthroplasty (line 64). In case of associated lesions of the lateral ligamentous complex, such lesions were treated during radial head prosthetic replacement; hence, no abnormal joint stability or biomechanics were detected in our sample.

Line 65: Please provide possible causes and conditions leading to primary radial head arthroplasty. In which way did these conditions influence joint biomechanics and possibly stress shielding?

R: In this study we included only primary implants, thus excluding revision arthroplasty. Primary arthroplasties were implanted for unreconstructible radial head fractures, malunions and symptomatic non-unions, as mentioned in the inclusion criteria. Considering that the etiopathogenesis of PRNR has yet to be clarified, we preferred to exclude conditions that are known, from the literature on other joints, to be factors that affect the integration of implants in order to reduce the influence of any possible bias.

Line 70: Please define in time manner the meaning of early. Please be also exact about the types of loosening.

R: “Early” means in the first 2 years after the implantation of a press-fit stem. We have reported in the manuscript (line 69) the types of loosening: “periprosthetic radiolucencies and ballooning osteolysis displaying lack of stem integration”.

Lines 71-72: Please define these technical errors.

R: “Technical errors” refers to errors in stem position (radial length) or alignment (prosthetic stems not in line with the radial shaft or non-properly oriented anatomical implants). We have added it in the text (line 72).

Line 74: Please describe the surgical management of these 29 cases of elbow terrible triad. How were the soft tissue injuries (tendons, ligaments etc.) assessed? In case of different management, did PRNR show any difference in severity or extension? Please mention these in the manuscript to prove the lack of heterogeneity in your sample or, otherwise, form subgroups excluding statistical errors.

R: Since the aim of this manuscript was to propose a new classification system, in the manuscript we only included patients that were classified and treated according to the current diagnostic and therapeutic algorithm. This study is based on mid- to long-term results, which were analyzed after definitive stabilization of PRNR in all the patients. The interesting topic pointed out by the reviewer in this comment might, however, represent the main aim of a future study. Other studies have confirmed that PRNR stabilizes after two years, regardless of the initial diagnosis and the type of treatment for associated lesions (Giannicola et al. 2023 BJJ) 

Lines 76-77: Please mention the surgical management of the elbow dislocation apart of radial head reconstruction. As previously mentioned, the target is to prove that your sample has low or no heterogeneity.

R: see previous comments

Lines 77-78: Please mention causes and degrees of elbow instability. How was this managed in each case? In which way did affect this the elbow biomechanics?

R: see previous comments

Line 78: Please mention causes and degrees of stiffness. How was this managed in each case? Which was the early postoperative rehabilitation protocol and which was the variance of range of motion? Could a limited range of motion affect bone density of radial head? 

R: see previous comments

Line 83-84: As you mentioned the least time frame for radiological assessment was 2 years. During this period, were no x-rays taken to rule out any other causes than stress shielding? Please clarify.

R: We considered, given the aim of this study, only X-rays performed after the surgical procedure and after a minimum of 24 months. Obviously, more X-rays were performed in this time interval for the clinical follow-up but these were not considered for the purposes of this study. Furthermore, as reported in those lines, the PRNR has a self-limiting nature (the first 2 years after the surgical procedure) and the maximum PRNR can be observed at 24 months. The literature contains another study on this topic (Giannicola et al. 2023 BJJ). Regarding the etiopathogenesis of PRNR, the biomechanical theory (stress shielding) is currently acknowledged to be the most likely.  (lines 277-306). 

Line 89: What did you mean rotatory stiffness? Is there a special subgroup of previous mentioned 8 cases of post-traumatic stiffness? Please clarify.

R: Rotatory stiffness means the limitation of the pronation and supination of the forearm. To visualize the four quadrants of the radial neck in these patients, X-rays were performed as described in these lines. For the purposes of this study, we did not divide patients into subgroups as we did not think this was necessary.

Line 96: Please mention exactly what is the meaning of pronounced differences. Moreover, was this range predefined or was assessed during the measurements?

R: We thank the reviewer for this comment. As reported in the statistical section, a preliminary study was performed to assess the reliability of the measurement methods and the reference intervals defined. The term sentence including the word “pronounced” has been changed to better clarify the preliminary study that was performed: “A third measurement was performed jointly by both operators if there were > 1mm differences between their measurements”(lines 110-111)

Lines 114-118: The rationale of cut-off points for proposed classification was fully and adequately presented. This is mandatory for any newly introduced classification.

R: We are grateful to the reviewer for the comment

Lines 130-131: Please mention in details the flowchart of this network review. Was it systematic review? If yes present it following PRISMA guidelines. In any other case you have to conduct a systematic review where pool data will be formed (if heterogeneity is low) and quantitive synthesis will be performed.

R: In the present study, a qualitative (narrative) review of the current literature on proximal radial neck resorption around RHA was performed and the results of the existing studies on this topic were reported. This was clarified at line 54.

Line 148: Regarding your results, you have to proof that your sample has no or minor heterogeneity and different causes (injuries or conditions) leading to radial head arthroplasty do not influence in different manner neck resorption. Quite recently, Giannicola et al (2023) has published the same results. Apart from the newly introduced classification, what is novel in this manuscript?

R: The main aim of this study was to propose a new quantitative classification, while the secondary aim was to perform a qualitative review of the literature. To be able to propose a new classification, it was necessary to validate it clinically first, and demonstrate its intra- and inter-observer reproducibility. The analysis of clinical results stratified for different PRNR grades was fundamental, even if in the study by Giannicola et al. (2023) the limited clinical relevance of this phenomenon had already been highlighted. Moreover, the statistical analysis about the correlations between PRNR grades and clinical results that was performed in this study is more thorough than the ones that were previously published by other authors. We believe that the analysis of correlations between the initial diagnosis and the severity of stress shielding deserves a dedicated new study.

Line 165: Please clarify what is the meaning of brackets in age, follow-up, MEPS, Q-DASH, P-ASES, SUP, PRON,EXT, FLEX.

R: We thank the reviewer for pointing out the error. The table has been corrected (line 198)

Line 212: Discussion section will be reassess after revisions and modification made in the previous sections according to above comments.

R: We believe that the discussion is well structured and does not need extensive adjustments, except for those made to address the specific comments previously made by the reviewers

Round 2

Reviewer 2 Report

Comments and Suggestions for Authors

Dear Authors,

All points raised were  adequately addressed and sufficient explanations provided by authors. As far as I am concerned, a new analysis with subgroups according to the primary indications for radial head arthroplasty could be possible.